# Attitudes toward COVID-19 and Other Vaccines: Comparing Parents to Other Adults, September 2022

**DOI:** 10.3390/vaccines11121735

**Published:** 2023-11-21

**Authors:** Matthew Z. Dudley, Holly B. Schuh, Michelle Goryn, Jana Shaw, Daniel A. Salmon

**Affiliations:** 1Institute for Vaccine Safety, Johns Hopkins Bloomberg School of Public Health, 615 N. Wolfe Street, Baltimore, MD 21205, USA; hschuh1@jhu.edu (H.B.S.); mgoryn1@jh.edu (M.G.); dsalmon1@jhu.edu (D.A.S.); 2Department of International Health, Johns Hopkins Bloomberg School of Public Health, Baltimore, MD 21205, USA; 3Department of Epidemiology, Johns Hopkins Bloomberg School of Public Health, Baltimore, MD 21205, USA; 4Department of Pediatrics, SUNY Upstate Medical University, Syracuse, NY 13210, USA; shawja@upstate.edu; 5Department of Health, Behavior and Society, Johns Hopkins Bloomberg School of Public Health, Baltimore, MD 21205, USA

**Keywords:** COVID-19, vaccine, vaccination, vaccine hesitancy, parents

## Abstract

Few analyses of COVID-19 vaccine attitudes also cover routine vaccines or focus on parents. In this cross-sectional study, we surveyed US adults in September 2022, immediately following the authorization of updated bivalent COVID-19 boosters for adults but before their authorization for children. The vaccine attitudes of parents were compared to other adults. Fewer parents were up-to-date on COVID-19 vaccines than other adults (54% vs. 67%), even after adjusting for age, education, and race/ethnicity (Adjusted Odds Ratio: 0.58; 95% Confidence Interval: 0.45–0.76). More parents had concerns about COVID-19 vaccines’ safety in children (67% vs. 58%; aOR: 1.59; 95%CI: 1.23–2.06) and vaccine ingredients (52% vs. 45%; aOR: 1.41; 95%CI: 1.09–1.81), and more parents perceived COVID-19 in children to be no worse than a cold or the flu (51% vs. 38%; aOR: 1.56; 95%CI: 1.22–2.01). Fewer parents supported COVID-19 vaccine school requirements (52% vs. 57%; aOR: 0.75; 95%CI: 0.58–0.97) and perceived high vaccine coverage among their friends (51% vs. 61%; aOR: 0.60; 95%CI: 0.46–0.78). However, three-quarters of parents intended their child to receive all routinely recommended vaccines, whereas only half of adults intended to receive all routinely recommended vaccines themselves. To improve parental informed vaccine decision-making, public health must ensure pediatric providers have updated resources to support their discussions of vaccine risks and benefits with their patients’ parents.

## 1. Introduction

Two COVID-19 mRNA vaccines have received Emergency Use Authorization (EUA) for children in the United States (US): Pfizer-BioNTech (Comirnaty) [1,2,3,4] and Moderna (Spikevax) [4,5]. Comirnaty was fully approved for adolescents 12 years and older in July 2022 [6]. Updated bivalent formulations of both were authorized in August 2022 to boost protection against circulating strains, initially just as booster doses for adults (Moderna and Pfizer) and adolescents (Pfizer) [7]. The EUAs for the updated bivalent vaccines were amended in October 2022 to include older children [8] and in April 2023 to include all children at least 6 months old, coinciding with an updated recommendation for all persons who have not yet done so to receive an updated vaccine [9]. Updated vaccine coverage remains poor, especially among children: as of 26 April 2023, just 5% of children 5–11 years old and 8% of adolescents 12–17 years old had received a bivalent dose, compared to 20% of adults [10]. The strongest predictors of parental intention to vaccinate their children against COVID-19 are the perceived benefits of the vaccine and previous acceptance of other routine vaccines for their children [11].

Although many parental vaccine concerns existed before the pandemic [12], and vaccine attitudes among US parents may have initially improved at the outset of the pandemic [13], overall, the pandemic seems to have had a negative impact on vaccine confidence. Vaccines unfortunately became a polarizing political issue during the pandemic [14], which, along with pandemic fatigue [15], has increased hesitancy to receive COVID-19 vaccines [16,17] and perhaps routine vaccines as well [18]. Frequent collection and review of representative survey data is needed to understand trends in vaccine hesitancy, especially if the drops in routine vaccine coverage from the early pandemic are to be regained [19,20].

We previously conducted a national panel survey in September 2021 [21], just before Pfizer’s EUA for children 5–11 years old [3], which included an analysis of the vaccine attitudes of parents compared to other adults [22]. About two-thirds (69%) of parents of children aged 2–17 years old had received at least one COVID-19 vaccine. Parents had lower odds of being vaccinated against COVID-19 than other adults, even after adjusting for associated sociodemographic characteristics such as age (aOR: 0.65; 95%CI: 0.49–0.87). Parents also had lower odds of having high confidence in routine vaccines (aOR: 0.76; 95%CI: 0.59–0.98). 

Herein, we describe findings from an analysis of another national panel survey conducted in September 2022 [23], a year after our previous survey [21,22], and immediately after the EUA of updated bivalent boosters for adults [7] but before their authorization for children [8]. The main focus of this analysis was to compare the vaccine attitudes of parents to other adults, both for COVID-19 and for other recommended vaccines. 

## 2. Materials and Methods

### 2.1. Study Design

A national panel survey was administered between 1 and 12 September 2022, in English and Spanish, using Ipsos KnowledgePanel [24], the largest probability-based online panel in the US. Ipsos uses address-based sampling techniques to recruit members to ensure the geodemographic composition of the panel mimics the adult US population. Stratified random selection, enrollment quotas, and survey weights ensured that the sociodemographic distribution of our sample remained representative of the adult US population even while oversampling Hispanic and Black respondents by 50% to increase the power to detect differences by race/ethnicity. We have successfully used Ipsos KnowledgePanel for related surveys previously [21,22,25,26]. More detail on the methodology of this survey is described in a previous publication [23], as well as in the STROBE checklist (Appendix A) [27]. The Institutional Review Board of the Johns Hopkins Bloomberg School of Public Health considered this study public health surveillance and not human subject research. 

### 2.2. Variables

One primary outcome of this survey was COVID-19 vaccination status. The survey began by asking respondents to identify themselves as either up-to-date (i.e., fully vaccinated and boosted); vaccinated but not up-to-date (e.g., have not gotten a booster yet); not vaccinated against COVID-19; or preferring not to say (a response only given by 3% of the sample and thus treated as missing). 

This survey also measured attitudes about vaccines for both children and adults, including constructs such as perceived susceptibility to and severity of vaccine-preventable diseases and the importance of COVID-19 vaccines. Among those not yet up-to-date on COVID-19 vaccines nor intending to get up-to-date as soon as possible, respondents identified their concerns and other reasons for not vaccinating. Confidence in sources of COVID-19 information, cumulative COVID-19 disease prevalence (ever having COVID-19 disease), and self-reported influenza vaccination were captured. Trust in the Centers for Disease Control and Prevention (CDC) was measured using a 14-item scale previously developed and validated [28]. A Cronbach’s alpha coefficient of 0.93 indicates the reliability of this scale. Further methodological details on the use of this scale are described in a previous publication [23].

Respondents were asked how many children (less than 18 years old) they had and the age of each child. Parents of at least one preteen 11–12 years old were given additional survey items pertaining to these children. Parents of children under 5 years old (who did not also have a child 11–12 years old) were given similar items pertaining to their child(ren) under 5 years old. These two age ranges were targeted to reflect the ages at which most vaccines are recommended according to the CDC schedule: most vaccines are given in the first five years of life, including vaccines required for kindergarten by law in many states (e.g., DTaP, polio, MMR, varicella, hepatitis B), though three vaccines (HPV, Tdap, meningococcal) are not recommended until 11 years of age [29]. Survey items measured parental intentions to get their children the vaccines recommended for their age group, confidence in the safety of these vaccines, self-efficacy (confidence they could get their child vaccinated), perceived vaccine knowledge, and specific vaccine concerns. Adults who were not parents of children 0–5 or 11–12 years old were given similar survey items but focused on adult vaccines recommended for themselves (since these adults were likely due the same number or more vaccines than their children), and were split by those 18–50 years old versus over 50 years old, again to reflect the vaccine schedule (e.g., the herpes zoster vaccine is only recommended for adults over 50) [30]. In essence, these four age groups of interest (parents of children 0–5, parents of preteens 11–12, adults 18–50, adults 50+) were made mutually exclusive, with each respondent only receiving the additional survey items for one of the four age groups, with priority given to the smallest group, to avoid redundancy and reduce survey length while maintaining power for precise estimates within each group. So, although all respondents received additional survey items, respondents who were parents of children both 0–5 years old and 11–12 years old were only given the additional survey items corresponding to the 11–12 age group, and adults 18–50 who were also parents of children 0–5 years old were only given the additional survey items corresponding to the 0–5 age group. 

Gender, age, race/ethnicity, education, income, employment status, metropolitan statistical area (MSA), region, and political affiliation were among the sociodemographic characteristics available for all respondents. Choices in survey content were influenced by the Health Belief Model [31] and the Social Ecological Model [32].

### 2.3. Measurement

The sample size for this survey was chosen to approximate the sample size of our previous related Knowledge Panel surveys, which were well-powered to demonstrate attitudinal associations with vaccine status and intentions [21,22,25,26]. Design weights were adjusted using a raking procedure to imitate the US adult population. Hispanic and Black respondents were oversampled to increase power for stratified analyses but down-weighted to reflect their proportion in the population (Table 1). Further details on this weighting technique have been published elsewhere [25].

Sociodemographic characteristics (Table 2) and vaccine attitudes (Table 3) were cross-tabulated against parent status, and vaccine attitudes were also cross-tabulated against oldest child age (comparing older to younger) (Appendix A). Odds ratios were calculated. General vaccine attitudes and safety concerns were cross-tabulated against general vaccine intentions (Table 4). Likert and other scale response options were dichotomized to reflect affirmative versus negative (e.g., agree vs. disagree, important vs. not important) to facilitate straightforward analysis and interpretation.

Standard errors for weighted proportions were calculated using Taylor-linearized variance estimation. *p*-values for cross-tabulations were calculated using the Pearson chi-squared proportion test (α = 0.05). Bivariate odds ratios were calculated using generalized logistic binomial regression with a logit link function. In Table 2, simple logistic regressions featured parent status as the dependent variable and other sociodemographic characteristics as independent variables. In Table 3, multiple logistic regressions featured affirmative survey responses as the dependent variable and parent status as the main independent variable, with the sociodemographic characteristics significantly associated with parent status in Table 2 included as additional independent variables to adjust for potential confounding. Data were analyzed using Stata statistical software (version 16) [33].

## 3. Results

### 3.1. Study Population and Survey Weighting

The survey was fielded among 5323 panel members. Of these, 2787 (52%) completed the survey, of which 2561 qualified for the study (based on eligibility criteria and survey quotas). Unweighted and weighted sociodemographic characteristics and vaccination status of the study population are presented in Table 1. Weighted data are generalizable to the adult population of the US. 

**Table 1 vaccines-11-01735-t001:** Sociodemographic characteristics and vaccination/disease status of the sample: unweighted and weighted.

Sociodemographic Characteristics	Unweighted N (%)	Weighted % ^a^	Sociodemographic Characteristics	Unweighted N (%)	Weighted % ^a^
All	2561 (100)		Household Annual Income		
Gender			<$50,000	847 (33.1)	29.9
Female	1274 (49.7)	51.4	$50,000–75,000	452 (17.6)	16.4
Male	1287 (50.3)	48.6	$75,000–100,000	346 (13.5)	13.1
Age (years)			$100,000–150,000	413 (16.1)	17.8
18–29	308 (12)	19.9	$150,000+	503 (19.6)	22.8
30–44	679 (26.5)	25.7	Political Affiliation		
45–59	669 (26.1)	23.9	Republican	517 (20.2)	25.3
≥60	905 (35.3)	30.5	Democrat	998 (39.1)	32.8
Education			Independent/Other	1039 (40.7)	41.9
<High School	223 (8.7)	9.6	Metropolitan Statistical Area Status		
High School	589 (23)	28.1	Non-Metro	276 (10.8)	13.4
Some College	757 (29.6)	27.1	Metro	2285 (89.2)	86.6
Bachelor’s or Higher	576 (22.5)	19.6	Parent Status		
Master’s or Higher	416 (16.2)	15.5	No Children <18	1794 (70.1)	71.4
Race/Ethnicity			At Least One Child <18	767 (29.9)	28.6
White, non-Hispanic	997 (38.9)	62.8	Influenza Vaccination ^b^		
Black, non-Hispanic	609 (23.8)	12.0	Not Vaccinated	1144 (45.2)	46.1
Hispanic	838 (32.7)	16.9	Vaccinated	1386 (54.8)	53.9
Other, non-Hispanic	117 (4.6)	8.3	COVID-19 Vaccination		
Region			Not Vaccinated	329 (13.3)	15.5
Northeast	415 (16.2)	17.3	Vaccinated (≥1 dose)	2140 (86.7)	84.5
Midwest	455 (17.8)	20.7	COVID-19 Disease		
South	1036 (40.5)	38.2	Never Had	1432 (56.7)	55.9
West	655 (25.6)	23.8	Ever Had	1093 (43.3)	44.1

^a^ Weights used iterative proportional fitting so that the sample represented US adults; Black and Hispanic respondents were weighted to adjust for their oversampling (to provide sufficient power for analyses stratified by race/ethnicity). ^b^ Respondents reported whether or not they had received an influenza vaccine within the past 12 months; these data were collected prior to the 2022–2023 influenza season, and so should reflect the 2021–2022 influenza season.

### 3.2. Parent Status

#### 3.2.1. Sociodemographic Characteristics Associated with Parent Status

Nearly 30% of the weighted sample had at least one child less than 18 years old (Table 2). Three sociodemographic factors were associated with being a parent: age, education, and race/ethnicity. 

**Table 2 vaccines-11-01735-t002:** Sociodemographic characteristics of parents (of children <18 years of age) versus other adults.

Survey Items	Total (%) ^a^ N = 2561	Parent (%) ^b^		*p*-Value ^c^	OR (95%CI) ^d^
		No	Yes		
All	100	71	29		
Gender				0.17	
Female	51	50	54		ref ^e^
Male	49	50	46		0.86 (0.70–1.07)
Age (years)				**<0.01**	
18–29	20	22	14		ref ^e^
30–44	26	14	55		6.37 (4.36–9.31)
45–59	24	23	25		1.71 (1.16–2.52)
60+	31	40	6		**0.23 (0.14–0.37)**
Education (attained)				**<0.01**	
<High School	10	8	12		ref ^e^
High School	28	30	24		**0.55 (0.37–0.82)**
Some College	27	28	24		**0.59 (0.40–0.87)**
Bachelor’s	20	20	19		**0.67 (0.45–0.99)**
Master’s or Higher	16	14	20		0.96 (0.64–1.45)
Race/Ethnicity				**<0.01**	
White, non-Hispanic	63	67	52		ref ^e^
Black, non-Hispanic	12	12	13		**1.46 (1.13–1.88)**
Hispanic	17	15	22		**1.95 (1.55–2.45)**
Other, non-Hispanic	8	7	12		**2.23 (1.42–3.50)**
Region				0.33	
Northeast	17	18	15		ref ^e^
Midwest	21	20	22		1.37 (0.97–1.95)
South	38	38	39		1.27 (0.93–1.73)
West	24	24	24		1.25 (0.90–1.75)
Household income				0.35	
<$50,000	30	31	27		ref ^e^
$50,000–75,000	16	17	16		1.07 (0.76–1.49)
$75,000–100,000	13	13	12		1.04 (0.74–1.47)
$100,000–150,000	18	17	19		1.22 (0.88–1.69)
$150,000+	23	22	26		1.32 (0.99–1.77)
Political affiliation				0.27	
Republican	25	26	24		ref ^e^
Democrat	33	34	31		0.98 (0.74–1.30)
Independent/Other	42	41	45		1.17 (0.89–1.55)
Metropolitan Statistical Area status				0.85	
Non-Metro	13	13	13		ref ^e^
Metro	87	87	87		1.03 (0.74–1.44)

^a^ Column percentages (of total sample N = 2561), weighted for national representativeness. ^b^ Column percentages (of parent status) (except for the first row, “All”, which is a row percentage), weighted for national representativeness. ^c^ Using the Pearson chi-square test; bold indicates statistical significance (*p* < 0.05). ^d^ Odds Ratio (95% Confidence Interval) of being a parent of at least one child <18 years of age and reporting the sociodemographic characteristics in each row; for example, survey respondents who were 60+ years old had 77% lower odds of being a parent of at least one child <18 years of age than survey respondents who were 18–29 years old; bold indicates statistical significance (*p* < 0.05). ^e^ Reference category for logistic regression of categorical variable.

#### 3.2.2. Vaccine and Disease Status Associated with Parent Status

Parents were less likely to be up-to-date on their COVID-19 vaccines than other adults (54% vs. 67%), even after adjusting for age, education, and race/ethnicity (Adjusted Odds Ratio: 0.58; 95% Confidence Interval: 0.45–0.76) (Table 3). Parents were also more likely to report ever having COVID-19 disease (54% vs. 40%; aOR: 1.50; 95%CI: 1.17–1.93). 

**Table 3 vaccines-11-01735-t003:** Vaccine attitudes and values of parents (of children <18 years of age) versus other adults.

Survey Items	Total (%) ^a^ N = 2561	Parent (%) ^b^		*p*-Value ^c^	aOR (95%CI) ^d^
		No	Yes		
**All**	100	71	29		
**Vaccination and Disease Status**					
Vaccinated against flu within the past year	54	55	52	0.28	1.13 (0.88–1.46)
Vaccinated against COVID-19 (at least one dose)	85	86	82	0.10	0.83 (0.56–1.22)
Up-to-date on COVID-19 vaccines	63	67	54	**<0.01**	**0.58 (0.45–0.76)**
Ever knowingly had COVID-19 disease	44	40	54	**<0.01**	**1.50 (1.17–1.93)**
**Scales**					
Trust in the Centers for Disease Control and Prevention (CDC)	68	68	67	0.60	0.91 (0.70–1.19)
**Confidence in sources of information about COVID-19**					
My doctor	86	86	86	0.94	1.08 (0.74–1.60)
My local or state health department	74	74	74	0.86	1.01 (0.75–1.36)
Scientists and doctors from the CDC	70	69	71	0.40	1.08 (0.81–1.44)
The Surgeon General	68	67	70	0.36	1.10 (0.82–1.46)
Scientists and doctors from universities	73	72	75	0.29	1.07 (0.79–1.45)
Dr. Anthony Fauci, National Institutes of Health	62	62	62	0.94	0.88 (0.67–1.14)
Dr. Rochelle Walensky, CDC Director	63	63	61	0.54	0.90 (0.69–1.17)
Dr. David Satcher, Morehouse School of Medicine, Former CDC Director and Surgeon General	62	62	62	0.98	0.94 (0.73–1.23)
My religious leader	31	30	32	0.32	1.23 (0.93–1.62)
Other non-medical people in my community that I trust	31	31	31	0.88	1.04 (0.77–1.40)
What I see on the news	37	40	31	**<0.01**	**0.68 (0.53–0.88)**
What I see on social media (Facebook, Twitter, etc.)	13	12	16	0.05	1.13 (0.78–1.63)
**Agreement with COVID-19 Likert Scale Items (for Adults)**					
I worry I may accidentally spread COVID-19 to my family members in the next six months.	30	29	35	**0.01**	0.99 (0.77–1.28)
I worry I may accidentally spread COVID-19 to my friends, neighbors, or co-workers in the next six months.	27	25	32	**<0.01**	1.05 (0.80–1.38)
If I get COVID-19, I think it will be severe.	14	14	15	0.46	1.00 (0.68–1.46)
COVID-19 vaccines are important to stopping the spread of infection in the US.	72	73	70	0.21	0.90 (0.68–1.20)
COVID-19 vaccines are important to helping the US get back to a normal life.	71	72	68	0.12	0.79 (0.60–1.05)
Most or all of my family members have gotten vaccinated against COVID-19.	65	66	63	0.18	0.84 (0.64–1.10)
Most or all of my friends have gotten vaccinated against COVID-19.	58	61	51	**<0.01**	**0.60 (0.46–0.78)**
If my main doctor were to recommend that I take the COVID-19 vaccine, I’d be likely to take it.	42	41	50	0.37	0.90 (0.80–1.02)
If a close family member were to recommend that I take the COVID-19 vaccine, I’d be likely to take it.	38	39	38	0.87	0.94 (0.61–1.43)
If my close friends were to recommend that I take the COVID-19 vaccine, I’d be likely to take it.	34	34	33	0.83	0.98 (0.63–1.54)
I feel knowledgeable about the COVID-19 vaccine.	76	78	72	**0.01**	0.79 (0.59–1.07)
I’d like to get more information on COVID-19 vaccines.	30	30	30	0.92	0.89 (0.68–1.17)
**Agreement with COVID-19 Likert Scale Items (for Children)**					
COVID-19 can be a serious disease for some children.	83	84	82	0.38	0.99 (0.69–1.41)
I am concerned about the safety of the COVID-19 vaccine in children.	61	58	67	**<0.01**	**1.59 (1.23–2.06)**
Vaccinating children against COVID-19 is important to end the pandemic and get back to normal.	65	66	61	0.06	0.73 (0.56–0.95)
It is better for children to develop immunity to COVID-19 by getting sick rather than by getting a shot.	37	33	45	**<0.01**	**1.56 (1.20–2.02)**
COVID-19 in children is no worse than a cold or the flu.	42	38	51	**<0.01**	**1.56 (1.22–2.01)**
I would support a requirement for children to be vaccinated against COVID-19 to attend school.	56	57	52	**0.05**	**0.75 (0.58–0.97)**
**Agreement with General Vaccine Likert Scale Items**					
I am confident in the safety of vaccines.	79	80	78	0.51	0.95 (0.68–1.33)
I do not trust a vaccine unless it has already been safely given to millions of other people.	46	44	52	**0.01**	1.26 (0.97–1.62)
I am concerned about some of the ingredients in vaccines.	47	45	52	**0.01**	**1.41 (1.09–1.81)**
Vaccine recommendations from the Centers for Disease Control and Prevention (CDC) are a good fit for me.	72	72	74	0.31	1.14 (0.85–1.52)
I am concerned that the government and drug companies experiment on people like me.	41	39	47	**0.01**	1.25 (0.96–1.62)
The benefits of vaccines are much bigger than their risks.	78	78	79	0.90	1.18 (0.86–1.61)

Red text indicates survey items reflecting negative vaccine attitudes. ^a^ Column percentages (of total sample N = 2561), weighted for national representativeness. ^b^ Column percentages (of parent status) (except for the first row, “All”, which is a row percentage), weighted for national representativeness. ^c^ Using the Pearson chi-square test; bold indicates statistical significance (*p* < 0.05). ^d^ Adjusted Odds Ratio (95% Confidence Interval) of reporting agreement with the survey item in each row comparing parents of at least one child <18 years of age to other adults, adjusted for the sociodemographic characteristics significantly associated with parental status in Table 2 (i.e., age, education, race/ethnicity); for example, parents of at least one child <18 years of age had 50% greater odds of having had COVID-19 disease than non-parents (after adjusting for age, education, and race/ethnicity); bold indicates statistical significance (*p* < 0.05).

#### 3.2.3. Vaccine Attitudes Associated with Parent Status

Parents were less likely to report that most of their friends had gotten vaccinated against COVID-19 than other adults (51% vs. 61%; aOR: 0.60; 95%CI: 0.46–0.78) or support a requirement for children to be vaccinated against COVID-19 to attend school (52% vs. 57%; aOR: 0.75; 95%CI: 0.58–0.97). Parents were more likely to report concerns about the safety of COVID-19 vaccines in children (67% vs. 58%; aOR: 1.59; 95%CI: 1.23–2.06) or vaccine ingredients (52% vs. 45%; aOR: 1.41; 95%CI: 1.09–1.81). Parents were also more likely to believe it better for children to develop immunity to COVID-19 by getting sick rather than by getting a shot (45% vs. 33%; aOR: 1.56; 95%CI: 1.20–2.02) or that COVID-19 in children is no worse than a cold or the flu (51% vs. 38%; aOR: 1.56; 95%CI: 1.22–2.01). 

### 3.3. Routine Vaccine Intentions, Safety Concerns, and Trust in CDC by Age Groups of Interest

#### 3.3.1. Routine Vaccine Intentions

Three-quarters (76%) of parents of preteens 11–12 years old intended for that child to receive all recommended vaccines in adolescence; 22% were unsure or intended to skip some adolescent vaccines, and 2% intended to skip all adolescent vaccines (Table 4). Four-fifths (80%) of parents of children under 5 years old (who did not also have a child 11–12 years old) intended for that child to receive all recommended vaccines in childhood; 18% were unsure or intended to skip some childhood vaccines, and 2% intended to skip all childhood vaccines. In contrast, two-fifths (41%) of adults 18–50 years old (who did not have a child 0–5 or 11–12 years old) intended to receive all vaccines recommended for young adults, while 41% were unsure or intended to skip some adult vaccines, and 18% intended to receive no adult vaccines; 54% of adults over 50 years old (who did not have a child 0–5 or 11–12 years old) intended to receive all vaccines recommended for older adults, while 38% were unsure or intended to skip some adult vaccines, and 9% intended to receive no adult vaccines.

**Table 4 vaccines-11-01735-t004:** Routine vaccine intentions and safety concerns of parents and other adults, stratified by age groups most relevant to vaccine schedule.

Survey Items (by Age Group) ^a^	Response	Total (%) ^b^	Intentions to Get Recommended Vaccines (%) ^c^			*p*-Value ^d^
			All	Some/Unsure	None	
**Parents of Children <5 Years (*n* = 231)**		**11**	**80**	**18**	**2**	
My child does not like needles.	Disagree	20	64	33	3	**0.02**
	Agree	80	84	15	2	
I am confident I can get my child vaccinated if I so choose—I know where to get vaccines I can afford, and I have transportation to get there.	Disagree	6	33	55	12	**<0.01**
	Agree	94	83	16	1	
I have most of the important information I need to make a decision about vaccinating my child.	Disagree	11	46	50	4	**<0.01**
	Agree	89	84	15	2	
I am confident that getting the recommended vaccines is safe for my child.	Disagree	13	17	72	11	**<0.01**
	Agree	87	89	10	1	
*Among those not confident in vaccine safety…*						
I worry that the ingredients in vaccines are unnatural or unsafe for my child.	Disagree	26	5	81	14	0.38
	Agree	74	21	69	9	
I worry about my child developing autism because of vaccines.	Disagree	65	19	75	6	0.55
	Agree	35	14	69	17	
I worry about the serious side effects of vaccines.	Disagree	28	4	86	9	0.29
	Agree	72	22	67	11	
I worry about my child getting too many vaccines at once.	Disagree	38	3	84	13	0.13
	Agree	62	26	65	9	
It is better to develop immunity by getting sick rather than by getting a shot.	Disagree	56	4	83	13	0.04
	Agree	44	33	59	8	
The flu vaccine is made with eggs, which my child is allergic to.	Disagree	79	22	68	10	0.42
	Agree	21	0	92	8	
The flu vaccine has thimerosal in it, which is dangerous.	Disagree	60	23	67	10	0.57
	Agree	40	9	81	10	
The flu vaccine will make my child sick with the flu.	Disagree	51	5	84	11	0.10
	Agree	49	30	62	8	
I worry about outbreaks of disease from vaccines.	Disagree	64	17	77	6	0.59
	Agree	36	17	66	17	
There is less disease nowadays because of better sanitation, not because of vaccines.	Disagree	73	23	67	10	0.33
	Agree	27	0	88	12	
I don’t see the point of vaccinating against diseases that are so rare.	Disagree	72	22	69	9	0.30
	Agree	28	5	79	16	
*Scales*						
Trust in the Centers for Disease Control and Prevention (CDC)	Low	35	67	30	3	**<0.01**
	High	65	87	11	1	
**Parents of Preteens 11–12 Years (*n* = 135)**		**5**	**76**	**22**	**2**	
My child does not like needles.	Disagree	23	62	33	5	**0.04**
	Agree	77	81	19	0	
I am confident I can get my child vaccinated if I so choose—I know where to get vaccines I can afford, and I have transportation to get there.	Disagree	6	0	87	13	**<0.01**
	Agree	94	81	18	1	
I have most of the important information I need to make a decision about vaccinating my child.	Disagree	5	0	85	15	**<0.01**
	Agree	95	80	19	1	
I am confident that getting the recommended vaccines is safe for my child.	Disagree	17	10	83	7	**<0.01**
	Agree	83	90	10	0	
*Among those not confident in vaccine safety…*						
I worry that the ingredients in vaccines are unnatural or unsafe for my child.	Disagree	24	29	62	10	0.13
	Agree	76	4	89	7	
I worry about the serious side effects of vaccines.	Disagree	33	22	71	7	0.33
	Agree	67	5	88	7	
I worry about my child getting too many vaccines at once.	Disagree	14	0	83	17	0.62
	Agree	86	12	82	6	
It is better to develop immunity by getting sick rather than by getting a shot.	Disagree	46	18	71	11	0.22
	Agree	54	3	92	4	
The flu vaccine is made with eggs, which my child is allergic to.	Disagree	88	10	83	7	0.94
	Agree	12	11	79	11	
The flu vaccine has thimerosal in it, which is dangerous.	Disagree	74	11	79	11	0.67
	Agree	26	13	87	0	
The flu vaccine will make my child sick with the flu.	Disagree	64	13	75	11	0.33
	Agree	36	5	95	0	
I worry about outbreaks of disease from vaccines.	Disagree	68	12	77	11	0.41
	Agree	32	6	94	0	
There is less disease nowadays because of better sanitation, not because of vaccines.	Disagree	44	3	81	17	0.09
	Agree	56	16	84	0	
I don’t see the point of vaccinating against diseases that are so rare.	Disagree	82	10	81	9	0.74
	Agree	18	11	89	0	
I am concerned about the HPV vaccine in particular.	Disagree	52	18	68	14	0.06
	Agree	48	1	99	0	
*Among those concerned about the HPV vaccine in particular…*						
I worry that the HPV vaccine is too new.	Disagree	48	0	100	100	0.47
	Agree	52	3	97	100	
I worry that the HPV vaccine could cause my child to develop chronic fatigue syndrome.	Disagree	63	0	100	100	0.31
	Agree	37	4	96	100	
I worry that the HPV vaccine could increase my child’s sexual activity.	Disagree	76	2	98	100	0.68
	Agree	24	0	100	100	
I worry that the HPV vaccine could cause my child to become infertile.	Disagree	65	0	100	100	0.30
	Agree	35	4	96	100	
*Scales*						
Trust in the Centers for Disease Control and Prevention (CDC)	Low	35	53	43	4	**<0.01**
	High	65	88	11	1	
**Adults 18–50 Years (*n* = 887)**		**52**	**41**	**41**	**18**	
I wish I better understood how vaccines actually work.	Disagree	56	41	39	20	0.10
	Agree	44	41	45	14	
I wish I better understood how vaccines are made and tested.	Disagree	41	43	38	19	0.24
	Agree	59	40	44	16	
I wish I better understood all the different types of vaccines.	Disagree	42	40	38	22	0.05
	Agree	58	42	43	14	
I am less than 25 years old and have never received an HPV vaccine.	Disagree	62	46	49	4	**0.01**
	Agree	38	30	47	23	
I do not like needles.	Disagree	46	42	40	19	0.67
	Agree	54	41	43	16	
I am confident I can get vaccinated if I so choose—I know where to get vaccines I can afford, and I have transportation to get there.	Disagree	10	12	57	31	**<0.01**
	Agree	90	44	40	16	
I have most of the important information I need to make a decision about vaccination.	Disagree	10	22	55	22	**0.01**
	Agree	90	43	40	17	
I am confident that getting the recommended vaccines is safe for me.	Disagree	24	6	46	48	**<0.01**
	Agree	76	52	40	8	
*Among those not confident in vaccine safety…*						
I worry that the ingredients in vaccines are unnatural or unsafe.	Disagree	22	7	54	39	0.57
	Agree	78	5	44	51	
I worry about the serious side effects of vaccines.	Disagree	17	6	60	34	0.38
	Agree	83	6	43	51	
It is better to develop immunity by getting sick rather than by getting a shot.	Disagree	30	14	57	28	**<0.01**
	Agree	70	2	41	57	
The flu vaccine is made with eggs, which I’m allergic to.	Disagree	94	5	47	48	0.19
	Agree	6	19	42	40	
The flu vaccine has thimerosal in it, which is dangerous.	Disagree	60	10	48	41	**0.04**
	Agree	40	0	42	58	
The flu vaccine will make me sick with the flu.	Disagree	50	8	47	44	0.34
	Agree	50	3	44	53	
I worry about outbreaks of disease from vaccines.	Disagree	49	8	50	42	0.32
	Agree	51	4	40	56	
There is less disease nowadays because of better sanitation, not because of vaccines.	Disagree	48	9	47	44	0.24
	Agree	52	2	44	54	
I don’t see the point of vaccinating against diseases that are so rare.	Disagree	58	8	58	35	**<0.01**
	Agree	42	3	28	70	
*Scales*						
Trust in the Centers for Disease Control and Prevention (CDC)	Low	33	21	44	35	**<0.01**
	High	67	52	40	8	
**Adults >50 Years (*n* = 1279)**		**48**	**54**	**38**	**9**	
I wish I better understood how vaccines actually work.	Disagree	53	54	35	11	**0.01**
	Agree	47	54	40	6	
I wish I better understood how vaccines are made and tested.	Disagree	44	61	29	11	**<0.01**
	Agree	56	49	44	7	
I wish I better understood all the different types of vaccines.	Disagree	42	56	33	11	**0.02**
	Agree	58	53	41	7	
I do not like needles.	Disagree	55	51	39	10	0.13
	Agree	45	57	36	7	
I am confident I can get vaccinated if I so choose—I know where to get vaccines I can afford, and I have transportation to get there.	Disagree	3	28	43	29	**<0.01**
	Agree	97	55	37	8	
I have most of the important information I need to make a decision about vaccination.	Disagree	7	21	69	10	**<0.01**
	Agree	93	56	35	9	
I am confident that getting the recommended vaccines is safe for me.	Disagree	23	3	61	36	**<0.01**
	Agree	77	69	30	1	
*Among those not confident in vaccine safety…*						
I worry that the ingredients in vaccines are unnatural or unsafe.	Disagree	14	1	68	31	0.44
	Agree	86	3	59	37	
I worry about the serious side effects of vaccines.	Disagree	9	3	72	24	0.47
	Agree	91	3	60	37	
It is better to develop immunity by getting sick rather than by getting a shot.	Disagree	31	8	74	18	**<0.01**
	Agree	69	1	54	45	
The flu vaccine is made with eggs, which I’m allergic to.	Disagree	93	2	61	38	**<0.01**
	Agree	7	20	62	18	
The flu vaccine has thimerosal in it, which is dangerous.	Disagree	60	2	64	34	0.17
	Agree	40	6	55	39	
The flu vaccine will make me sick with the flu.	Disagree	66	3	67	30	0.08
	Agree	34	3	49	47	
I worry about outbreaks of disease from vaccines.	Disagree	64	2	65	33	0.22
	Agree	36	4	52	43	
There is less disease nowadays because of better sanitation, not because of vaccines.	Disagree	49	5	61	34	0.28
	Agree	51	1	61	37	
I don’t see the point of vaccinating against diseases that are so rare.	Disagree	53	5	65	29	**0.02**
	Agree	47	1	57	43	
*Scales*						
Trust in the Centers for Disease Control and Prevention (CDC)	Low	25	16	59	25	**<0.01**
	High	75	69	29	2	

^a^ Each survey respondent (N = 2561) only received the above survey items for one of the four age groups, with priority given to the smallest group, to avoid redundancy and an unnecessarily long survey while maintaining power for precise estimates within each group. So, although all respondents received additional survey items, respondents who were parents of children both 0–5 years old and 11–12 years old were only given the additional survey items corresponding to the 11–12 age group, and adults 18–50 who were also parents of children 0–5 years old were only given the additional survey items corresponding to the 0–5 age group. The exception is the scale items that went to the entire sample. ^b^ Column percentages of the total sample for each age group, and of the total in the corresponding (mutually exclusive) age group for each survey item, weighted for national representativeness. ^c^ Row percentages of intentions to get vaccines recommended for each (mutually exclusive) age group by agreement with survey item, weighted for national representativeness. For parents of children <5 years of age, we asked their intentions to get those children the vaccines recommended for them at this age; for parents of children 11–12 years of age, we asked their intentions to get those children the vaccines recommended for them at this age; for adults 18–50, we asked their intentions to get themselves the vaccines recommended for them at this age (e.g., annual flu and Tdap every 10 years); for adults >50, we asked their intentions to get themselves the vaccines recommended for them at this age. ^d^ Using the Pearson chi-square test; bold indicates statistical significance (*p* < 0.05).

#### 3.3.2. Vaccine Safety Concerns

Most parents of preteens 11–12 years old (83%) and other parents of children under 5 years old (87%) were confident in the safety of the vaccines recommended for their child; about three-quarters of other adults (76% of adults 18–50, 77% of adults over 50) were confident in the safety of the vaccines recommended for themselves. Confidence in routine vaccine safety was strongly correlated with intending to receive routine vaccines among all four age groups (*p* < 0.01). 

The most prevalent specific vaccine concerns among parents of preteens 11–12 years old who were not confident in vaccine safety included worrying about their child getting too many vaccines at once (86%), their child not liking needles (77%), worrying that the ingredients in vaccines are unnatural or unsafe (76%), and worrying about the serious side effects of vaccines (67%). About half of parents of preteens 11–12 years old who were not confident in vaccine safety believed it better to develop immunity by getting sick rather than by getting a shot (54%) and that there is less disease nowadays because of better sanitation, not because of vaccines (56%). The most prevalent vaccine concerns among other parents of children 0–5 years old who were not confident in vaccine safety were the same: not liking needles (80%), ingredients (74%), serious side effects (72%), and getting too many vaccines at once (62%). Nearly half of other parents of children 0–5 years old who were not confident in vaccine safety believed that the flu vaccine would make their child sick with the flu (49%) and that it is better to develop immunity by getting sick rather than by getting a shot (44%). 

The most prevalent vaccine concerns among other adults who were not confident in vaccine safety included: serious side effects (83% of adults 18–50, 91% of adults over 50), ingredients (78% of adults 18–50, 86% of adults over 50), and preferring to develop immunity by getting sick rather than by vaccinating (70% of adults 18–50, 69% of adults over 50). About half of other adults did not like needles (54% of adults 18–50, 45% of adults over 50), believed modern disease control to be due to sanitation, not vaccines (52% of adults 18–50, 51% of adults over 50), and did not see the point of vaccinating against rare diseases (42% of adults 18–50, 47% of adults over 50). About half of other adults 18–50 and about one-third of other adults over 50 worried about outbreaks of disease from vaccines (51% and 36%, respectively) and believed flu vaccines would give them the flu (50% and 34%, respectively). Two-fifths (40%) of other adults worried about the dangers of flu vaccines containing thimerosal. Very few other adults were concerned about allergic reactions to eggs in flu vaccines (6% of adults 18–50, 7% of adults over 50).

#### 3.3.3. Trust in CDC

About two-thirds of parents of preteens 11–12 years old (65%), other parents of children 0–5 years old (65%), and other adults 18–50 years old (67%) had high trust in the CDC. Three-quarters (75%) of adults over 50 years old had high trust in CDC. Trust in CDC was strongly correlated with intending to receive routine vaccines among all four age groups (*p* < 0.01). 

## 4. Discussion

Parents of children were less likely to be up-to-date on their COVID-19 vaccines than other adults and more likely to report ever having COVID-19 disease. Parents were more likely to report concerns about vaccine safety and ingredients and perceive COVID-19 in children to be no worse than a cold or the flu. Parents were less likely to support COVID-19 vaccine school requirements and to perceive high vaccine coverage among their friends. However, more than three-quarters of parents of children aged 0–5 or 11–12 (ages at which most routine vaccines are due) intended for their child to receive all recommended vaccines for their age, whereas only about half of adults without children these ages intended to receive all recommended vaccines themselves. 

These findings mostly align with data from our previous surveys. In our September 2021 survey, parents were less likely to be vaccinated against COVID-19 and more likely to report concerns about vaccine safety and perceive low COVID-19 disease severity than other adults, though parents were no more likely than other adults to report having had COVID-19 disease at that point [22]. We hypothesized at the time that parents may have had more preexisting vaccine hesitancy prior to the pandemic than non-parents, perhaps due to a greater flow of vaccine misinformation among parent social circles or negative experiences vaccinating their children. Parent concerns reflect those of the general US population, albeit often more prevalent [21,25,34].

Some of these concerns are also understandably and unsurprisingly higher among parents, given the circumstances of the COVID-19 pandemic and parents’ experiences during it. For example, parents likely perceive COVID-19 to be more similar to routine winter colds than other adults because COVID-19 is typically less severe among children and young adults than older adults [10], and so parents’ personal experiences with COVID-19 (whether among themselves or their children) may have been less severe than the experiences of other adults. Additionally, parents likely perceive lower vaccine coverage among their friends because many of their friends are also parents, among whom vaccine coverage is lower. COVID-19 vaccines have also been studied more thoroughly among adults than among children, with the initial clinical trials and EUAs focusing on persons at least 16 years of age [1,5,7,35,36], potentially explaining the greater frequency of vaccine safety concerns among parents compared to other adults. 

These findings also highlight an interesting contrariety: parents are less confident in vaccines than other adults, and COVID-19 vaccine coverage is much higher among adults than children [10], yet routine (non-COVID-19) vaccine coverage among adults [37] is far lower than among children [38]. For example, among children born during 2018 or 2019, primary series completion by 2 years of age was above 90% for most common childhood vaccines, such as hepatitis B, polio, measles/mumps/rubella (MMR), and varicella, and 64% had received at least two doses of influenza vaccine [39]. In contrast, only 14% of US adults had received all age-appropriate vaccines in 2018, with coverage rates of 46% for influenza, 63% for tetanus, 69% for pneumococcal (among adults over 65), and 24% for herpes zoster (among adults over 50) [40]. However, this apparent contradiction is likely explained by convenience and expectation; most children see their pediatrician regularly and are given routine vaccines as part of the long-established standard of care, whereas many adults (especially younger adults) do not regularly see a doctor and thus often must seek out their own vaccination independently. 

### 4.1. Implications

Vaccinated parents are far more likely to intend to vaccinate their children than unvaccinated parents [22]. Thus, improving vaccine coverage and attitudes among parents should also increase the likelihood that these parents will vaccinate their children. Messaging and outreach should support decision-making by emphasizing vaccine effectiveness and addressing concerns with sensitivity and clarity. Since healthcare providers are the most common and credible source of vaccine information for parents [41,42], especially as opposed to government or pharmaceutical companies, public health should prioritize supplying useful resources to healthcare providers to aid in their support of the vaccine decision-making of their patients (or their patients’ parents). Such resources should be updated regularly to reflect new circulating strains, authorized vaccines, and/or public safety concerns.

### 4.2. Strengths and Limitations

The CDC now considers everyone 6 years and older to be “up-to-date” if they have received an updated COVID-19 vaccine (children 6 months to 5 years may need multiple doses to be up to date, but at least one dose must be the updated COVID-19 vaccine) [9,43]. However, when this survey was administered in September 2022, “up-to-date” was defined as fully vaccinated (with a primary series) and boosted. No specification was made regarding whether the booster must be the most recent version, as the bivalent booster was not yet widely available, having just been authorized [7]. Thus, we are unable to differentiate between booster versions in our data and assume that our measure of “up-to-date” refers largely to the now obsolete monovalent booster. However, our survey was also well-timed to capture the proportion of US parents and other adults who had received a monovalent booster dose just before it was replaced by the updated bivalent version. 

Another limitation of our study is its reliance on data from one point in time rather than over an extended period. Our data are also subject to the limitations of self-reporting. However, most analyses of COVID-19 vaccine attitudes do not cover routine vaccines, nor have they focused on parents in particular, as ours does. Furthermore, many other analyses of COVID-19 vaccine attitudes have not been subjected to peer review, and their internal and external validity varies widely. A strength of this analysis is its use of high-quality data from a well-established nationally representative panel.

## 5. Conclusions

Immunization programs must reemphasize and sustain efforts to support parents as they make vaccine decisions both for themselves and their children. The public health community should ensure pediatric providers have the resources needed to discuss the risks and benefits of vaccines with their patients’ parents, especially as new vaccines are authorized, and recommendations are updated. 

## Data Availability

Deidentified individual participant data will not be made available. The data are not publicly available to protect participant confidentiality.

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
