# Peer review of "Attitudes toward COVID-19 and Other Vaccines: Comparing Parents to Other Adults, September 2022"

_vaccines, 2023, doi:10.3390/vaccines11121735_

Round 1

Reviewer 1 Report

Comments and Suggestions for Authors

In this work, the Authors updated (September 2022) the results obtained from a previous survey conducted during the COVID-19 pandemic (September 2021) on attitudes of US population toward COVID-19 and routine vaccinations, comparing parents and non-parents individuals.

The results of this survey are interesting. Please, consider my comments below:

-       All the Tables in the Methods should be moved to the Results section

-       Table 4 – first and second columns are not readable, please separate the questions with lines, like Table 2, otherwise the answers will get confused

-       Table 2, you can leave the black lines just to separate the main sections (gender, age, education, and so on), while delete lines for the subsections

-       Line 291: a year prior, may be one year before?

-       Lines 336-342: you can add other limitations of the study clearly, i.e. no specification if the booster was the most recent version, data refer to the monovalent booster dose that became obsolete, others if any.

-       negative vaccine attitudes highlighted in table 3 could be discussed, i.e. potentially dangerous ingredients in vaccines formulations, VAERS data, emergency use of COVID-19 vaccines and concern that the government and drug companies “experiment on people like me”, and so on. As you stated, most of the parents/adults showed to be not “no-vax” people, and your results on concerns and attitudes might have had evidence-based reasons that must be taken into account.

Reviewer 2 Report

Comments and Suggestions for Authors

The statistical presentation and analyses are poor.  N's should be available for every table, not just percentages, especially since sweeping conclusions are being drawn from relatively small samples, e.g., unknown proportion of the 30% of respondents whose kids were ages 5 or younger among the total give for people who had kids under age 18.  The value of the adjusted Odds Ratios aren't clear and there is no explanation of what variables were used for adjustment.

The statistics are problematic.  for example, see table 4.  The difference between parents concerns about autism risk among parents under age 5 shows a difference of 75% to none, and shows a p-value of 0.55 - odd for such a stark difference.  Supplementary table also indicates a problem with the statistical analyses.  See for example, the data comparing vaccine attitudes of parents with younger versus older children.  For the question, “most of all of my friends have gotten vaccinated against COVID-19” you claim a meaningful adjusted odds ratio supporting a big difference when the percentages shown are 50% vs 51%.  This can’t be right.  Made me wonder even more about the correctness or attention to detail in the anlayses.

Reviewer 3 Report

Comments and Suggestions for Authors

Dear author(s),

Thank you for your esteemed efforts in enhancing our collective knowledge about parental vaccine attitudes, especially in the context of COVID-19 booster vaccination.

The study was well designed and conducted and the manuscript was well-written too. I have only few minor comments below:

1. Conducting an analytical cross-sectional study to grab an updated picture of parental attitudes towards COVID-19 vaccine booster doses was indeed a good idea, but it remained unclear why the key denominator was (parent vs non-parent). This sounds as if you are attempting to describe all parent populations as a whole without enough emphasis on the correlates of vaccination attitudes.

2. All tables need to be moved to Results section rather than Methods.

3. The Methods section needs to be re-structured according to the STROBE guidelines.

4. STROBE guidelines need to be cited, and the STROBE checklist should be added as a supplementary file.

5. There is no information on sample size calculation.

6. There is no information on how the instrument (questionnaire) used in this study was tested for its validity and reliability.

7. How did you ensure equivalency between English and Spanish versions?

8. The Introduction section does not address the various determinants of parental vaccination attitudes sufficiently. It was only limited to the authors' previous work, which is relevant and interesting, but definitely not enough to synthesise an appealing narrative that justifies further research.

9. Line 339 - 342: this limitation does not sound clear to me.

10. The Discussion section will benefit from having three subheadings:
- Strengths
- Limitations
- Implications

Sincerely,

Round 2

Reviewer 2 Report

Comments and Suggestions for Authors

Nice paper and much improved revision.  Below I list a few areas that didn't make sense to me or were unclear, including one glaring statistical issue:

 Abstract states “more parents perceived low COVID-19 disease severity.”  Not sure what this means.  Same phraseology is used in several places.  Does it mean that they thought  if their kids got COVID, it would not be severe, or is this a broader statement about COVID variants that were circulating in September 2022?  Especially since this statement is in sharp contrast to table 3 item: COVID-19 can be a serious disease for some children where 82% of parents indicated they agreed with this statement.

Introduction should be updated to reflect full approval of Pfizer vaccine for children.  The manuscript gives the impression that none of the COVID-19 vaccines have received full approval for use in children.

Section 2.1 Study design last sentence.  Is this an implication of the IRB position that no informed consent was sought?  Good description of panel

Page 3 lines, 120-121, describes the supplementary surveys for 4 categories of interest, but has a qualifier that I don’t understand:  “with priority given to the smallest group, to avoid redundancy and reduce survey length while maintaining power for precise estimates within each group.”  Did everyone receive the appropriate supplementary questions or only some groups?

Table 3 has some odd statistics, e.g., I worry I may accidentally spread COVID to my family members, and also coworkers.   They describe the differences between parents and non-parents as statistically significantly different, but the odds ratios are 0.99 and 1.05 and both confidence intervals include one, so this can’t be correct.

 Table 4 – I found it particularly interesting that most parents said they have most of the information they need to make a decision about vaccinating their child and reassuring that the autism risk perception was not higher.
